# An Epigenetic Role of Mitochondria in Cancer

**DOI:** 10.3390/cells11162518

**Published:** 2022-08-13

**Authors:** Yu’e Liu, Chao Chen, Xinye Wang, Yihong Sun, Jin Zhang, Juxiang Chen, Yufeng Shi

**Affiliations:** 1Tongji University Cancer Center, Shanghai Tenth People’s Hospital of Tongji University, School of Medicine, Tongji University, Shanghai 200092, China; 2Department of Neurosurgery, Changhai Hospital, Second Military Medical University, 168 Changhai Road, Shanghai 200433, China; 3Department of Pharmacology and Toxicology, University of Mississippi Medical Center, Jackson, MS 39216, USA; 4Clinical Center for Brain and Spinal Cord Research, Tongji University, Shanghai 200092, China

**Keywords:** mitochondria, epigenetics, cancer, metabolism

## Abstract

Mitochondria are not only the main energy supplier but are also the cell metabolic center regulating multiple key metaborates that play pivotal roles in epigenetics regulation. These metabolites include acetyl-CoA, α-ketoglutarate (α-KG), S-adenosyl methionine (SAM), NAD^+^, and O-linked beta-N-acetylglucosamine (O-GlcNAc), which are the main substrates for DNA methylation and histone post-translation modifications, essential for gene transcriptional regulation and cell fate determination. Tumorigenesis is attributed to many factors, including gene mutations and tumor microenvironment. Mitochondria and epigenetics play essential roles in tumor initiation, evolution, metastasis, and recurrence. Targeting mitochondrial metabolism and epigenetics are promising therapeutic strategies for tumor treatment. In this review, we summarize the roles of mitochondria in key metabolites required for epigenetics modification and in cell fate regulation and discuss the current strategy in cancer therapies via targeting epigenetic modifiers and related enzymes in metabolic regulation. This review is an important contribution to the understanding of the current metabolic-epigenetic-tumorigenesis concept.

## 1. Introduction

Mitochondria in eukaryotic cells likely evolved from bacteria by endosymbiosis [1,2], and they are composed of two separate and functionally distinct membranes (the outer membrane and the inner membrane), the intermembrane space, and matrix compartments. Mitochondria also have a circular genome called mitochondrial DNA (mtDNA) [3]. The mitochondrial genome encompasses thousands of copies of the mtDNA and more than one thousand nuclear DNA (nDNA)-encoded genes [4]. Mitochondria form a dynamic and interconnected network that is intimately integrated with other cellular compartments such as nuclear, endoplasmic reticulum (ER), and plasma [5,6,7]. Mitochondria are not only the center for energy metabolism, but also the headquarters for different catabolic and anabolic processes, calcium fluxes, and various signaling pathways [8,9,10,11]. During these processes, many intermediates are generated, which are critical for cell anabolic metabolism and cellular signaling, as well as for providing substrates required for epigenetic modification [12]. Metabolites that are essential for epigenetic modification include acetyl coenzyme A (Acetyl-CoA), oxidized nicotinamide adenine dinucleotide (NAD^+^), α-ketoglutarate (α-KG), succinate, etc., and they perform diverse functions, including contributing to gene transcription modulation and cell fate determination.

The concept of epigenetics can be traced back to 1942. Waddington described the divergence between genotype and phenotype during development [13]. Now, epigenetics commonly refers to heritable alterations in gene expression without DNA sequence changing [14,15]. Epigenetic modification mainly includes DNA and histone modifications, which lead to nucleosome remodeling and gene transcription alternations. There are many types of epigenetic modifications, but all are tightly regulated by the specific enzymes and the corresponding substrates required [16]. The expression of enzymes is determined by the cell linage and state, while the substrates are highly dependent on the cell metabolism, which is tightly regulated by extracellular signaling and available nutrition [17,18,19]. These epigenetic modifications are also strictly dependent on the cellular energy status [20]. Metabolites generated in the tricarboxylic acid cycle (TCA) in mitochondria or related metabolism are crucial for epigenetics, which links the relation among mitochondria, gene expression, and cell fate dictation. We briefly explored the function of several important metabolites in basic epigenetics such as DNA methylation and histone modification. The aberration of epigenetics exerts a significant impact on cellular function, thus leading to the dysregulation of gene expression and finally resulting in various diseases, such as cancers [21].

Oncogene overactivation and tumor suppressor inactivation transform somatic cells into cancer cells [22,23,24]. This transformation can take place during tissue regeneration and can be accelerated under infections, toxins, or other metabolic influences that cause mutations [25,26]. After transformation, additional gene mutation, gene expression alteration, and change in tumor microenvironment drive tumor evolution [27].

Historically, the notion of cancer stem cells (CSCs) is adopted from somatic stem cells, and it refers to a subpopulation of cancer cells with relatively slow cycling but that can regenerate the entire tumor after treatment. CSCs originate from either differentiated cells or tissue-resident stem cells [28]. CSC was first discovered as the leukemia-initiating cell in human acute myeloid leukemia in 1994 [29]. After that, CSCs were also found in tumors in the brain, breast, colon, liver, etc. [30,31,32,33,34]. These CSCs drive tumor initiation and tumor malignant growth and are often responsible for drug resistance and relapses. Unlike normal somatic stem cells, CSCs evolve with tumor evolution, which results from additional gene expression alternations and tumor microenvironment changes. CSCs can propagate tumors and promote tumor progression compared with the non-tumorigenic cells within the bulk tumor [35].

Metabolic and epigenetic reprogramming is essential for tumor initiation, evolution, and metastasis [36,37]. Epigenetics dictates the cell fate of both embryonic cells and somatic cells in differentiation and development [38], and different epigenetic regulations on gene expression also affect tumorigenesis [39,40]. A multitude of epigenetic mechanisms, including DNA methylation and post-translational modification of histones, contribute to diversity within tumors [41,42]. Epigenetic deregulation participates in the earliest stages of tumor initiation and has been increasingly recognized as a hallmark of cancer [43,44]. The deregulation of various epigenetic pathways contributes to tumorigenesis, especially concerning the maintenance and survival of CSCs [35]. For example, DNA hypermethylation has been associated with the silencing of tumor suppressor genes as well as differentiation genes in various cancers [45].

Considering the critical role of mitochondria in cellular metabolism and the essential role of epigenetics in tumorigenesis, it is recognized that the interplay between mitochondria metabolism and epigenetic enzymes plays a critical role in tumor initiation and evolution [46,47,48,49]. The metabolites such as acetyl-CoA, α-KG, S-adenosyl methionine (SAM), NAD^+^, and O-linked beta-N-acetylglucosamine (O-GlcNAc) play crucial roles in cell fate determination by regulating epigenetic modifications and gene expression.

Tumorigenesis affected by both genetic and non-genetic determinants constitutes a major source of therapeutic resistance. Targeting epigenetics as well as mitochondrial metabolism has been demonstrated as a promising therapeutic strategy to inhibit tumor growth. Mitochondrial and epigenetic targeting cancer therapy reduces tumor malignancy and often exhibits synergistic effects when combined with classical cytotoxic cancer therapy.

## 2. Mitochondria Function in Epigenetic Regulation

Mitochondria are the energy and metabolic center in the cell, which are essential for catabolic and anabolic metabolism. Mitochondrial function is tightly regulated by the expression of the mitochondrial genome as well as the surrounding microenvironment, both of which determine the cellular level of acetyl-CoA, α-KG, SAM, NAD^+^, and O-GlcNAc required for epigenetic modification [50,51]. Epigenetic modification is a genetic regulation model that is independent of DNA sequence and plays an important role in establishing and maintaining specific gene expression. DNA methylation, histone post-translational modification, and chromatin remodeling are the major forms of epigenetic modifications [52,53]. Epigenetic alterations are the main mechanisms in development and tumorigenesis. Mitochondria, as the main energy supplier, play a pivotal role in the regulation of epigenetics [54].

### 2.1. Mitochondria and DNA Methylation

Methylation is the only known modification that occurs on DNA, RNA, and proteins [55].

DNA methylation is a major form of DNA modification in the mammalian genome, which mainly occurs at cytosine in a dinucleotide CpG [56,57,58]. DNA methyltransferases (DNMTs) are responsible for the establishment and maintenance of methylation patterns [59]. They transfer the methyl group donated from SAM to cytosine and generate 5-methylcytosine (5mC) [60] (Figure 1). DNMT1 is responsible for maintaining the existing DNA methylation during DNA replication, and DNMT3A and DNMT3B are responsible for establishing de novo DNA methylation patterns [61]. The pattern of DNA methylation in the genome changes is a dynamic process regulated by both de novo DNA methylation and demethylation [62].

The demethylation of DNA requires the function of the ten-eleven translocation (TET) enzyme. TETs belong to the 2-oxoglutarate-dependent dioxygenases (2-OGDO) family and their activity requires coenzyme factors such as Fe^2+^ and α-KG [63,64]. They catalyze the oxidation of 5-methylcytosine (5mC) to 5-hydroxymethylcytosine (5hmC), 5-formylcytosine (5fC), and 5-carboxylcytosine (5caC) [65,66]. 5hmc resulted from the hydroxylation of methylcytosine and hydroxylation is involved in more than 90% reduction in cytosine methylation. 5hmC is not only an intermediate in the DNA demethylation process but also an independent epigenetic marker [67,68,69,70].

#### 2.1.1. SAM Source and Mitochondrial Regulation of SAM

SAM is a significant biological sulfonium compound that participates in a variety of biochemical processes. SAM is synthesized through the reaction of methionine with ATP catalyzed by methionine adenosyltransferase (MAT). This reaction mainly occurs in the cytosol through the one-carbon (1C) metabolism pathway that encompasses both the folate and methionine cycles [71]. Since SAM is synthesized in the cytosol [72], it has to be transported into mitochondria by the mitochondrial SAM carrier, and it is converted into S-adenosylhomocysteine (SAH) in the methylation reaction of DNA, RNA, and proteins [73]. SAH is further hydrolyzed in a reversible reaction by SAH hydrolase (SAHH) to give rise to adenosine and homocysteine. SAH is a competitive inhibitor of the methylation process, therefore both the increase of SAH and the decrease of SAM or SAM/SAH ratio inhibit the transmethylation reactions [74,75].

SAM is utilized in the cell by three metabolic pathways: transmethylation, transsulfuration, and polyamine synthesis [72]. It is the universal and sole methyl donor to DNMTs. The methyl group from SAM can be transferred to a variety of substrates, including nucleic acids, proteins, phospholipids, biologic amines, and other small molecules [74].

SAM levels are regulated by the level of methionine that is an essential amino acid in one-carbon metabolism, and methionine deficiency reduces liver SAM levels and H3K4 trimethylation drastically [76]. 1C units can be generated from serine by cytosolic or mitochondrial folate metabolism, and the loss of mitochondrial pathway renders cells dependent on extracellular serine to make 1C units [77].

#### 2.1.2. α-KG Source and Its Function in DNA Methylation

α-KG is generated in cell cytoplasm via glycolysis and in mitochondria from isocitrate catalyzed by isocitrate dehydrogenase (IDH) [78]. It is a membrane-impermeable crucial metabolite in the TCA cycle, contributing to the oxidation of nutrients such as amino acids, glucose, and fatty acids and it then provides energy for cell processes [79]. The level of α-KG changes upon fasting, exercise, and aging [80,81].

α-KG possesses a variety of physiological functions, besides its function in the TCA cycle in ATP production, and it reacts with ammonia and is converted into glutamate and glutamine. It can be metabolized into succinate, carbon dioxide (CO_2_), and water, and it also eliminates H_2_O_2_ by this process, acting as an antioxidant [79,82].

α-KG is an important metabolic intermediate and cofactor for several chromatin-modifying enzymes [83]. The family of α-KG-dependent dioxygenases includes Jumonji C-domain (JmjC) lysine demethylases, TET DNA cytosine-oxidizing enzymes, and prolyl hydroxylases (PHDs) [84]. In terms of DNA methylation, α-KG is an essential cofactor for αKG-dependent dioxygenases and the TET family of DNA demethylases [65,66,85]. α-KG is dramatically required and is increased for the activation of DNA methylation of the PRDM16 promoter during the early stage of adipogenesis [86]. In AML, decreased intracellular α-KG levels cause DNA hypermethylation through altering TET activity [87].

The level of α-KG is regulated by other metabolites. Fumarate and succinate, the intermediates of the TCA cycle and substrates of fumarate hydratase (FH) and succinate dehydrogenase (SDH), are competitive inhibitors of α-KG-dependent dioxygenases [88,89,90] (Figure 1). The accumulation of fumarate and succinate caused by mutation of enzymes inhibits α-KG-dependent enzymes, including TETs [91], thus leading to the global decreases in 5hmC and inactivation of FH and SDH and causing the loss of 5hmC [90,92,93]. Both fumarate and succinate accumulation are involved in glycolytic pathways by the inhibition of TET1 and TET3 [90]. FH-mediated TET inhibition promotes epithelial-mesenchymal transitions (EMT)-related transcription factors and enhances cell migration [88]. Mutations in isocitrate dehydrogenase 1 (IDH1), another TCA enzyme, lead to the accumulation of 2-hydroxyglutarate (2-HG), and 2-HG acts as a competitive inhibitor of α-KG-dependent enzymes, including TETs [66,93]. IDH mutations not only generate 2-HG but also result in hypermethylation of tumor suppressor genes [94]. Non-specific α-KG analogue dimethyloxallyl glycine (DMOG) has been used as a TET inhibitor, which blocks multiple α-KG-dependent dioxygenases [64].

#### 2.1.3. Mitochondrial ROS and DNA Methylation

Reactive oxygen species (ROS) is regarded as the byproduct of oxygen consumption and cellular metabolism [95]. Mitochondria and NADPH oxidases (NOX) are two major contributors to endogenous ROS in tumor cells [96]. ROS plays an important role in physiology and the pathophysiology of aerobic life [97]. Excessive or inappropriately localized ROS damages cell growth. Higher levels of ROS contribute to tumor development through both genetic and epigenetic mechanisms [98,99,100]. ROS also acts as signaling molecules in DNA damage, thus promoting genetic instability and tumorigenesis. ROS-induced oxidative stress is closely related to both hypermethylation of tumor suppressor gene promoter regions and global hypomethylation.

ROS can function as the catalyst of DNA methylation by facilitating the transfer of methyl group to cytosine such as the formation of large complexes containing DNMT1/DNMT3B or increasing the expression of DNMTs, thus leading to hypermethylation of genes [101,102,103]. DNMT expression can be upregulated by superoxide anion, which is a form of ROS. The elevation of superoxide anion upregulates DNMT1 and DNMT3B and inhibits superoxide anion, thus leading to a decrease in DNMT upregulation and DNA methylation. In colon cancer, H_2_O_2_ reduces the expression of RUNX1 by elevating methylation at its promoter [104]. H_2_O_2_ also induces the upregulation of DNMT1 and HDAC1, thus leading to gene silence via histone deacetylation and promoter methylation [105,106]. In a brain function study, it was found that 5mC levels seemed to be increased and the 5hmC levels decreased coupled with increased ROS in the adult cerebellum, which suggested the promotion role of ROS in DNA methylation [107].

Epigenetic modifications within the mammalian nuclear genome include DNA methylation (5mC) and hydroxymethylation (5hmC) or DNA demethylation [65,108]. DNA hydroxymethylation can occur as a result of oxidative stress or the action of TET1 [65,108]. Oxidative stress has been suggested to activate TET via the increased production of α-KG and leads to an increase in 5hmC levels [109]. As the TET enzymes that convert 5mC to 5hmC may respond to oxidative stress, oxidative stress leads to the demethylation of genes. The redox-linked alterations to TET function can be reflected by the addition of ascorbate (vitamin C) to cellular 5hmC levels. Vitamin C is an antioxidant that can scavenge the primary ROS, and vitamin C treatment leads to an increase in 5hmC levels; thus, demethylation was promoted [110,111,112].

Besides ROS, oxygen availability modulates both DNMT’s and TETs’ enzymatic activity, and hypoxia induces hypermethylation by reducing TET activity [104,113,114]. DNA hypermethylation and hypoxia are well-recognized cancer hallmarks. TET belongs to the subfamily of 2OG oxygenases, which also act as an oxygen sensor [115]. Tumor hypoxia directly reduces TET activity, thus causing a 5hmC decrease predominantly at gene promoters and enhancers. Reduced TET activity leads to an accumulation of 5mC, thus decreasing the expression of associated genes. 5hmC is reduced under hypoxia in a dose-dependent manner, with decreasing concentrations of purified TET1 and TET2 [113].

#### 2.1.4. Mitochondria Dysfunction and DNA Methylation

Mitochondrial dysfunction invokes mitochondria to nucleus responses [116]. It reduces oxygen and SAM-CH_3_ production in the cell, and the unbalanced redox homeostasis leads to perturbed methylation of nuclear DNA [117]. The mtDNA copy number is one of the signals by which mitochondria affect nDNA methylation patterns; changes in mtDNA copy number, often observed in cancer, induce changes in nDNA methylation [118]. Cells with depleted mitochondria (rho0 cell) result in the aberrant methylation of promoter CpG islands which were previously unmethylated in the parental cells, and the repletion of mitochondria back into rho0 cells reestablishes methylation partially to their original parental state, indicating the crucial role of mitochondria for DNA methylation [118,119]. 

The presence of mtDNA methylation is controversial [120,121,122]. In 2011, Shock et al. reported that mammalian mitochondria have mtDNA cytosine methylation; mitochondrial DNA methyltransferase (mtDNMT1) binds to mtDNA in the mitochondrial matrix; mtDNMT1 is upregulated by nuclear respiratory factor 1 (NRF1), PGC1-α, and the loss of p53. The expression of mtDNMT1 asymmetrically affects the expression of transcripts of mtDNA, and thus affects the function of mitochondria [123,124]. However, this discovery was overturned in 2021 by Lacopo Bicci et al. [125], and they found that there is no CpG methylation in human mtDNA via whole genome bisulfite sequencing (WGBS) for studying the methylation of mtDNA.

The demethylase TETs’ activity requires coenzyme factors such as Fe2^+^ and α-KG. IDH mutation leads to the increase of succinate, fumarate, and 2-HG, and they inhibit TET activity. DNMTs transfer the methyl group from SAM to substrates. SAM is regulated by 1C unit metabolism. ROS can function as the catalysts of DNA methylation by facilitating the transfer of methyl group to cytosine and by adding vitamin C; an antioxidant can scavenge the ROS and leads to the increase in 5hmC. The oxidative stress also activates TET via increasing α-KG and finally leads to an increase in 5hmC.

### 2.2. Mitochondria and RNA Methylation

Over 100 different types of RNA modification have been identified. RNA methylation, especially the N6-methyladenine (m^6^A) modification, is the most abundant RNA modification confirmed in viruses, yeast, and mammals [126,127]. m^6^A modification was established by pioneering studies in 1974 [128,129]. Around 15,000 human genes have m^6^A modification, which is a widespread regulatory mechanism that controls gene expression by affecting the splicing, stability, and translation efficiency of the modified mRNA molecules in diverse physiological processes [127,130,131,132]. m^6^A RNA modification affects cell division, immune homeostasis, and biological rhythm. The dysregulation of m^6^A modification has been suggested to drive human cancers [133].

#### 2.2.1. The Effect of Mitochondria-Associated Metabolites on RNA Methylation

RNA methyltransferases, also named RNA methylation writers such as METTL3-METTL14, move the methyl group from SAM to mRNA, and the demethylases FTO and ALKBH5 remove the RNA methylation [134]. RNA methylation is decoded by proteins such as YTHDF1 and YTHDF2 [130]. FTO and ALKBH5 belong to the α-KG-dependent dioxygenase family [135]. The α-ketoglutarate dehydrogenase (OGDH) is the first rate-limiting enzyme in the TCA cycle, and ALKBH5 promotes viral replication metabolically by upregulating OGDH [136]. Furthermore, the enzymatic activity of FTO is inhibited in IDH mutation AML cells, thus leading to an increase in m^6^A levels [137,138].

In addition, several mitochondrial proteins also regulate RNA methylation. For example, mitochondrial RNA methyltransferase METTL8 facilitates 3-methyl-cytidine (m^3^C) methylation of the mt-tRNA and thus promotes mitochondrial ETC activity [139].

#### 2.2.2. The Roles of RNA Methylation in Mitochondria

Mitochondrial function is closely associated with RNA methylation, including both mRNA and tRNA methylation. Mitochondria possess specific translation machinery for the synthesis of proteins in ETC, and tRNA modification is indispensable during mitochondrial translation and the OXPHOS process, which require folate-dependent tRNA methylation [140]. A recent study indicates that mitochondrial tRNA methylation is necessary for tumor metastasis due to its critical role in mitochondria gene translation [141]. Moreover, tRNA methyltransferase 10C (TRMT10C), catalyzing the mitochondrial ND5 mRNA at N1-methyladenosine (m^1^A) site, promotes cancer cell metabolism by reducing the mitochondrial ribonuclease P protein 1 (MRPP1) in mitochondria and inducing protein instability and mt-tRNA processing [142]. In short, RNA methylation plays pivotal roles in mitochondrial synthesis and function.

### 2.3. Mitochondria and Histone Modifications

Besides DNA methylation, mitochondrial metabolism also plays a pivotal role in the modifications of histone tails [143]. Chromatin structure and function are regulated by post-translational modifications (PTMs) of histones, such as histone methylation, acetylation, succinylation, ubiquitylation, and SUMOylation [144]. By altering interactions between nucleosome components, including histones and DNA, histone PTMs affect chromatin structures. Histone modifications are both highly dynamic and tightly regulated by the histone modification enzymes as well as their substrates, which are tightly regulated by mitochondria [145] (Figure 2).

#### 2.3.1. Mitochondria and Histone Acetylation

**a**.
**Concept and enzymes of histone acetylation**


Histone acetylation usually occurs on the lysine residues of histone tails. Acetyl groups are transferred from acetyl-CoA by histone acetyltransferases to the ε-amino group of lysine residues. Acetylation in histone tails generally reduces interaction between histones and DNA and increases accessibility to transcription factors [146,147]. Three major types of acetylation have been described based on the acetylation site in the protein, including: (a) N-terminal acetylation that is commonly named N-ter acetylation or Nα-acetylation; (b) lysine acetylation that is commonly named K-acetylation (KAc) or Nε-acetylation; (c) O-acetylation [148]. Histone acetylation is one of the key epigenetic modifications regulated by histone acetyltransferases (HATs) and histone deacetylases (HDACs) [149,150,151,152]. HATs catalyze the transfer of an acetyl group from acetyl-CoA to the ε-NH2 group of lysine residues in proteins while HDACs remove it [153]. HDACs are organized in four classes depending on sequence identity and domain organization. Class I (HDAC1, 2, 3, and 8), class II (HDAC4, 5, 6, 7, 9, and 10), and class IV (HDAC11) are zinc-dependent HDACs, while class III (SIRT1–7) deacetylases are NAD^+^-dependent [154]. The NAD^+^-dependent deacetylases sirtuins consist of seven members (SIRT1 to SIRT7) with different subcellular locations: SIRT1 and SIRT2 are in the nucleus, and cytosol SIRT3, SIRT4, and SRIT5 are in mitochondria, while SIRT6 and SIRT7 are in the nuclear location. Sirtuins play important roles in inflammation, cell growth, circadian rhythm, energy metabolism, neuronal function, and stress resistance [155,156,157].

**b**.
**Function of histone acetylation**


Histone acetylation loosens contacts between histones and DNA, and it masks the positive charges present on the lysine residues, reducing the affinity between histones and negatively charged DNA, facilitating the recruitment of transcriptional co-activators, and making transcription factor binding sites more accessible, thereby enhancing gene transcription [149]. HDACs remove the acetyl groups, tightening the interaction between DNA and histones, thus repressing transcription. Therefore, HATs play the transcription-promoting role via histone acetylation while HDACs conduct the transcription-delimiting function through deacetylation [158]. HATs function as both oncogenes and tumor suppressors in cancer. The deregulation of these enzymes by genetic or epigenetic alterations accompanied by defects in gene transcription has been implicated in oncogenesis [159]. Aberrations in histone modifications occur frequently in cancer, including changes in their levels and distribution at gene promoters, gene coding regions, repetitive DNA sequences, and other genomic elements [160]. Low histone acetylation levels are associated with a poorer prognosis of cancer patients due to increased risk of tumor recurrence and/or decreased survival probability [161].

**c**.
**Mitochondria regulate histone acetylation**


The TCA intermediate metabolites in mitochondria such as acetyl-CoA and NAD^+^ regulate the nuclear epigenome as the substrate of acetylation and a cofactor of deacetylation, respectively [54,162]. Other co-factors, including flavin adenine dinucleotide (FAD) and α-KG are closely associated with the above process, and the biosynthesis of the metabolic coenzymes depends on intracellular ATP levels and mitochondrial function [20,163,164].

Histone acetylation levels are strictly dependent on energy status. Reduced energy supplies not only reduce acetyl-CoA levels but also lead to the oxidation of NADH to NAD^+^. Since NAD^+^ but not NADH is a substrate for histone deacetylases—the sirtuins, energy deficiency activates the sirtuins to deacetylate the histones, thus increasing the positive charge of the chromatin proteins, causing chromatin condensation, and finally suppressing transcription, replication, growth, and proliferation [20]. When the energy supply is sufficient, acetyl-CoA is upregulated, and ATP and acetyl-CoA phosphorylate and acetylate chromatin, thus opening the nDNA for transcription and replication, and histone acetylation will be increased. When cells are in starvation or calories are limited, acetyl-CoA is required to be oxidized in the mitochondria for ATP synthesis. Chromatin phosphorylation and acetylation are lost, gene expression is suppressed, and the nucleo-cytosolic acetyl-CoA will be reduced [20].

Mitochondria are required to maintain histone acetylation (H3K27ac) in human colon cancer cells after inhibiting the electron transport chain (ETC) in mitochondria or depleting mitochondrial DNA, and histone acetylation is reduced accordingly. Pan-histone acetylation and H3K27ac marked acetylation are reduced in ρ0 DLD-1 cells compared with their complete parental cells [165]. In human glioblastoma tumors, histone acetylation is generally upregulated, and ROS pathways have negative correlations with H2B acetylation [166]. In addition, the increased level of lactic acid generated from glycolysis contributes to the higher local acidic pH to promote histone deacetylation and favors an aggressive and pro-metastatic phenotype of cancer cells [167,168].

Genes in mitochondria also affect histone acetylation. For example, C1qbp, which is primarily located in the mitochondrial matrix but can also be found at the cell surface, cytosol, and nucleus, is required for mitochondria oxidative phosphorylation (OXPHOS) as an RNA-binding protein to facilitate mitochondrial translation [169,170,171]. There is a reduction of acetyl-CoA in C1qbp-deficient CD8^+^ T cells and a significant decrease of H3K27ac in effector genes [172,173].

**d**.
**NAD^+^ function in histone acetylation**


Nicotinamide adenine dinucleotide exists in two forms inside the cell (in the cytoplasm, nucleus, Golgi, mitochondria, and other peroxisomes), oxidized form NAD^+^ and reduced form NADH. NAD^+^ can be synthesized de novo from tryptophan or through the recycling of its precursors. In de novo pathway, it is synthesized through the Preiss–Handler pathway from nicotinic acid (NA) or via salvage of the NAD^+^ precursors such as nicotinamide mononucleotide (NMN), nicotinamide riboside (NR), reduced nicotinamide mononucleotide (NMNH), or reduced nicotinamide riboside (NRH). Besides the Preiss–Handler pathway, NAD^+^ salvage also occurs through the recycling of other precursors such as nicotinamide (NAM) [174]. It plays an important role in intermediary metabolism, especially in numerous oxidation/reduction reactions [174].

NAD^+^ is required for over 500 enzymatic reactions and plays key roles in the regulation of almost all major biological processes [175]. It is used as a substrate by several families of enzymes (the so-called NAD^+^ consumers): sirtuins (SIRTs), poly (ADP-ribose) polymerases (PARPs) and cyclic ADP-ribose (cADPr) synthases [176,177,178,179], and all of them generate NAM as a result of NAD^+^ utilization.

The proper acetylation level within cells is modulated by the balanced activities of HAT and HDAC, and HDACs are regulated by fluctuations in NAD^+^ concentrations [180,181]. NAD^+^ served as a catalytic cofactor for HDAC III (sirtuins)-mediated histone deacetylation reactions [181,182]. Sirtuin family is NAD^+^-dependent protein lysine deacetylases, playing important roles in various physiological functions, from energy metabolism to stress response [183]. The elevated level of NAD^+^ has been demonstrated to activate Sirt1 and reduce acetylation [184].

**e**.
**Acetyl-CoA function in histone acetylation**


Acetyl-CoA is a metabolite derived from several pathways such as glycolysis, fatty acid oxidation, and amino acid catabolism [185,186,187]. The generation of acetyl-CoA via glycolysis is achieved by pyruvates. Pyruvates are synthesized from glycolysis. When oxygen is sufficient in normal cells, they traverse the mitochondria via the voltage-dependent anion channel (VDAC) or porin and the impermeable inner mitochondrial membrane with the aid of two mitochondrial pyruvate carriers (i.e., MPC1 and MPC2). Then, they are converted to acetyl-CoA to fuel the TCA cycle for efficient energy supply. In the mitochondria, acetyl-CoA is catalyzed by citrate synthase with oxaloacetate to yield citrate, and citrate can be transported to the cytosol and cleaved by ATP-citrate lyase (ACL) to regenerate acetyl-CoA and oxaloacetate [188]. Mammalian cells generate acetyl-CoA in mitochondria with a complex of nucleus-encoded proteins known as pyruvate dehydrogenase complex (PDC) [189]. PDC can be translocated to the nucleus generating acetyl-CoA in the nucleus independent of mitochondria. The knockdown of nuclear PDC in isolated functional nuclei decreased the de novo synthesis of acetyl-CoA and acetylation of core histones [190]. The dynamic translocation of mitochondrial PDC provides a pathway for nuclear acetyl-CoA synthesis required for histone acetylation and epigenetic regulation [190].

Acetyl-CoA controls key cellular processes, including energy metabolism, mitosis, and autophagy [186]. Mitochondrial acetyl-CoA is used to synthesize ketone bodies as an alternative fuel. Besides roles in ATP production, acetyl-CoA also represents a key node in metabolism due to its intersection with many metabolic pathways and transformations [187]. Glycolytic pyruvate enters the TCA cycle through acetyl-CoA, and acetyl-CoA is a key precursor of lipid synthesis and the sole donor of the acetyl groups for acetylation [186,191].

Acetyl-CoA, acting as a substrate for HATs, is crucial for histone acetylation. Alteration of pools of intracellular acetyl-CoA manipulates histone acetylation [164,192], and lower nucleo-cytosolic acetyl-CoA limits histone acetylation and cell proliferation. Many protein acetylation modifications are indeed regulated by acetyl-CoA availability [193,194].

Histone acetylation is also affected by modulating enzymes that regulate acetyl-CoA syntheses such as acetate-dependent acetyl-CoA synthetase 2 (ACSS2) and citrate-dependent ACL [164,195]. ACSS2 and ACL are two principal enzymes that generate Acetyl-CoA for histone acetylation [186]. ACSS2 directly regulates histone acetylation in neurons and spatial memory in mammals [196].

#### 2.3.2. Mitochondria and Histone Methylation

**a**.
**Histone methylation**


The addition of methyl group from SAM on lysine or arginine amino acid residue in histone, usually on H3 or H4, is marked as histone methylation [197]. Lysine residues undergo single, double, or trimethylation modification, while arginine residues only undergo single and double methylation modification (symmetrically or asymmetrically), acting as active or repressive marks of gene expression. Common lysine methylations on H3, including H3K4, H3K79, and H3K36, are markers of active transcription, whereas H3K20, H3K9, and K3K27 are associated with suppressive transcription [198]. Histone methylation, as one of the most important histone marks, plays a critical role in gene expression, cell cycle, genome stability, and chromosome maintenance [199].

**b**.
**Enzymes of histone methylation**


Histone methylations are catalyzed by histone methyltransferase (HMTs) enzymes [199]. HMTs include lysine methyltransferases (KMTs) such as SET-domain-containing enzymes and Dot1-like proteins and arginine methyltransferases such as protein arginine methyltransferase (PRMTs) [200,201,202]. Histone lysine and arginine methylation can be removed by histone demethylases (HDMs) [203,204]. Lysine demethyltransferases (KDMs) have been clustered into two families: the Jumonji domain depending on iron and oxoglutarate as cofactors and amine oxidases relying on FAD as a cofactor [163,205,206,207]. The two families are also named Jumonji-domain histone demethylase (JHDM, Jumonji domain containing family) and LSD (lysine-specific demethylase). LSD1 can specifically remove the single and double methylation modification of histones H3K4 and H3K9, while Jumonji family can remove the modification of lysine trimethylation. Sites on H3K 4, 9, 27, 36, 79, and H4K20 can be methylated, among which histones H3K4 and H3K9 are two common modification sites.

**c**.
**Mitochondrial metabolites and histone methylation**


Metabolites in mitochondria regulate histone methylation. Besides serving as a cofactor for TET-mediated DNA demethylation, α-KG also serves as a cofactor for JHDM family and HDM activity to modify epigenetic marks [208,209]. IDH1-mediated α-KG modulates trimethylation of H3K4 in the promoters of genes associated with brown adipogenesis [210]. Ectopic expression of SDH inhibits histone demethylation and hydroxylation of 5mC, suggesting that defective SDH contributes to tumor growth positively [93]. Nicotinamide N-methyltransferase (NNMT) is a metabolic enzyme that increases mitochondrial function and reduces oxidative stress [211], meanwhile catalyzing the transfer of methyl moiety from SAM to nicotinamide. Cancer cells with the overexpression of NNMT exhibit alterations in SAM level and histone methylation, and with the procurement of a more aggressive phenotype [212].

The histone methylation and acetylation are not separated modification processes, and they are dynamically interactive. For instance, histone H3K27ac distinguishes active enhancers from inactive/poised enhancer elements containing H3K4me1 alone [213].

#### 2.3.3. Mitochondria and Histone Succinylation

**a**.
**Concept of histone succinylation**


Lysine succinylation (Ksucc) is a newly discovered histone post-translational modification that changes the chemical environment of histones similar to other acylation modifications [214,215,216]. Histone lysine succinylation has been considered as an intermediate reaction of succinyl group (-CO-CH_2_-CH_2_-COOH) transfer from succinyl-CoA to homoserine [217]. It refers to the process of covalently binding a succinyl group to the ε-amine of lysine with the enzymatic or non-enzymatic mechanisms [218,219]. It brings more significant structural changes in space than other modifications like methylation and acetylation since succinylation changes the charge state of lysine residues to a high degree [214,219]. Most lysine succinylation is located in mitochondria, followed by the cytoplasm and the nucleus.

Lysine succinylation was identified in *E coli* proteins first and later in mammalian proteins, which indicates that lysine succinylation is a highly evolved conserved marker in eukaryotes. [214,216]. Lysine succinylation generally accumulates at transcriptional start sites to modulate gene expression. Histone succinylation at H3K79 contributes to transcription and tumor development [220].

**b**.
**Enzymes of histone succinylation**


The writers and erasers that are responsible for catalyzing the succinylation and desuccinylation are still under investigation, but accumulated evidence has indicated that succinylation is modulated by acetyltransferase and deacetylases [221]. Lysine acetyltransferase 2A (KAT2A, also named GCN5) has been identified as a histone succinyltransferase that catalyzes the succinylation of histone H3K79 [220]. Histone acetyltransferase 1 (HAT1) succinylated histone H3 at K122, which could provide epigenetic and gene expression regulation in cancer cells [222]. The NAD^+^-dependent deacetylase SIRT5 located in mitochondria has been recognized as desuccinylase, and succinylation is sensitive to both the increase and decrease of SIRT5 [216,223,224,225]. Another deacetylase SIRT7 is an eraser of succinylation of histone H3K122 for modulating DNA damage response and genome stability [226].

**c**.
**Mitochondrial metabolites and histone succinylation**


Succinyl-CoA is the succinyl donor produced in the cytoplasm, nucleus, or in the mitochondrial tricarboxylic acid (TCA) [215]. The concentration of succinyl-CoA produced in TCA regulates the level of histone succinylation [215].

Intracellular succinyl-CoA is mainly synthesized in mitochondria and is produced from α-KG, succinyl-carnitine, and succinate [227]. Succinyl group output from mitochondria to histone lysine can be achieved through permeable nuclear pores [228]. The oxoglutarate carrier SLC25A11 transports α-KG from mitochondria [229,230], and α-KG is converted to succinyl coenzyme A in the nucleus under the catalyzation of α-KG dehydrogenase (α-KGDH). Succinyl-CoA provides materials for GCN5, and then GCN5 activates succinyl-transferase to succinylate a histone lysine [220].

The dysfunction of mitochondria affects histone succinylation, and defective TCA cycle metabolism results in histone hypersuccinylation and non-histone hypersuccinylation associated with transcriptional responses [231].

#### 2.3.4. Mitochondria and Histone O-GlcNAcylation

**a**.
**Concept of O-GlcNAcylation**


O-linked-N-acetylglucosaminylation (O-GlcNAcylation) is a post-translational modification by glycosylation that links a single GlcNAc molecule to the serine/threonine site on proteins by an O-linked β-glycosidic bond [232]. *O*-GlcNAcylation is a key integrator of cellular nutritional status and occurs in the nucleus, cytoplasm, and mitochondrion. It participates in the process of nutrient sensing and metabolism and plays crucial roles in physiology and physiopathology [233,234].

**b**.
**Enzymes of O-GlcNAcylation**


The addition and removal of monosaccharides are regulated by O-GlcNAc transferase (OGT) and O-GlcNAcase (OGA), respectively. O-GlcNAcylation is highly enriched on histone and other proteins in nuclear pore complexes [235,236,237,238]. OGT has three isoforms: nucleocytoplasmic OGT, mitochondrial OGT (mOGT), and the shortest form of OGT. The mOGT isoform is targeted to the mitochondrial matrix and is pro-apoptotic [239,240], and it tends to accumulate in the mitochondrial inner membrane. OGT utilizes UDP-GlcNAc to catalyze the addition of O-GlcNAc to target proteins.

Histone O-GlcNAcylation that occurs on H2A and H2B may be linked to histone exchange, and core histones H3 and H4 can also be modified with O-GlcNAc [241,242,243,244]. Three sites of O-GlcNAcylation were mapped by tandem mass spectrometry on histone H2A Thr101, histone H2B Ser36, and histone H4 Ser47, respectively [245]. The modification sites of *O*-GlcNAcylation on histone are distinct from H3 and H4 tail modification, which is linked to the regulation of gene expression more directly. O-GlcNAcylation on H3 Ser10 site is mitosis-specific and the O-GlcNAc is conditionally phosphorylated as well, which points out the crucial role of OGT in modifying histone H3 during mitosis [246,247,248]. The role of OGT in gene transcription regulation is probably associated with polycomb repression, which is linked with the trimethylation of histone H3 at Lys-27 [249,250,251].

**c**.
**Mitochondria and O-GlcNAcylation**


The synthesis of UDP-GlcNAc is from the hexosamine biosynthetic pathway (HBP), which is a series of enzymatic reactions requiring key metabolites, including glucose, glutamine, ATP, and acetyl-CoA [252,253]. The pool of UDP-GlcNAc available for *O*-GlcNAcylation is influenced by many factors, including metabolic flux. Glutamine is one of these key metabolites, and the regulated entry of glutamine may be influenced by the synthesis of UDP-GlcNAc [254]. Acetyl-CoA, generated from the conversion of pyruvate via the mitochondrial pyruvate dehydrogenase complex during the oxidation of glucose and the β-oxidation of fatty acids, is one of the precursors of UDP-GlcNAc [255].

Mitochondria are considered the primary source of ATP production in the cell. Adenosine-monophosphate-activated protein kinase (AMPK) is a sensor of energy status whose activity is controlled by ATP/AMP ratio, and it serves as a guardian of metabolism and mitochondrial homeostasis. AMPK has been identified as a regulator of H2B Ser112 O-GlcNAcylation, and the activation of AMPK decreases H2B Ser112 O-GlcNAcylation and H2B K120 ubiquitination [244].

**d**.
**O-GlcNAcylation and other modifications**


Besides the functions in O-GlcNAcylation, OGT also binds to the TET class of DNA cytosine demethylases that may serve as 5-methylcytosine erasers [256,257,258]. O-GlcNAcylation may promote DNA demethylation by interacting with members of the TET family proteins, thus favoring gene transcription. O-GlcNAcylation influences other PTMs of histones indirectly and treating cells with glucosamine to increase O-GlcNAcylation is associated with decreased phosphorylation of histone H3 Ser10 [248]. O-GlcNAcylation at Ser642 of histone deacetylase 4 (HDAC4) is cardioprotective in diabetes mellitus [259]. In addition, the O-GlcNAcylation of HDAC1 overexpressed in hepatocellular carcinoma (HCC) promotes cancer progression, thus inhibiting O-GlcNAcylation of HDAC1, repressing the progression of HCC [260].

#### 2.3.5. Mitochondria and Histone Lactylation

**a**.
**Concept and function of histone lactylation**


Lactylation is a novel histone acylation code proposed in 2019 [261]. Twenty-eight lactylation sites on core histones in human and mouse cells have been identified [261]. Lactylation of histone lysine residues derived from lactate directly modulates gene transcription from chromatin in a p53-dependent and p300-mediated manner [262]. Lactate is a critical energy source for mitochondrial respiration and a primary circulating TCA substrate [263].

Lactylation plays important roles in inflammation and cancer. In BMDMs, the treatment with lactate derived from tumor cells drives tumor-associated macrophages to an M2-like phenotype, and this polarization increases depending on the treating time [261,264]. Furthermore, histone lactylation regulates gene expression during M1 macrophage polarization. It also promotes oncogenesis by facilitating m^6^A reader protein YEHDF2 expression in ocular melanoma [265]. In addition, the elevation of exogenous lactate rescued the defect of B cell adapter for PI3K (BCAP)-deficient BMDMs to enhance histone lactylation [266]. In lung fibrosis, histone lactylation at the promoter regions of representative genes was increased in lactate-treated macrophages.

Histone lactylation has been demonstrated as an epigenetic mark of the glycolytic switch. Lactate and acetyl-CoA are the key metabolites to bridge the glycolytic switch with two distinct epigenetic coding programs: lactylation and acetylation. Both can be generated from glycolysis. In malignant tumor cells, pyruvates are converted to lactate, and the glycolytic switch is adapted to meet the higher demands of fast cell growth and proliferation [267]. Lactate accumulation in tumor microenvironment leads to reprogrammed cell metabolism and induces a glycolytic switch or the Warburg effect [24].

**b**.
**Regulation of histone lactylation**


Histone lysine lactylation is specially modulated by lactate level. Lactate is a major energy source for mitochondrial respiration and the major gluconeogenic precursor [268], and it has major influences on energy substrate partitioning. Lactate secreted by fermenting cells can be oxidized or used as a gluconeogenic substrate by other cells and tissues, and the relation between lactate and mitochondria has been extensively reviewed elsewhere [269,270]. The accumulation of lactate by exercise affects the expression of regulatory enzymes of glycolysis and mitochondrial biogenesis [271]. Lactate is the fulcrum of metabolic regulation in vivo [268].

Lactate metabolism influences epigenetics under diverse cellular conditions. Hypoxia induces the production of lactate by glycolysis and stimulates histone lactylation [261]. Mitochondrial inhibitors or hypoxia with increased lactate production amplify lysine lactylation [272]. Histone lactylation and acetylation compete for epigenetic modification of lysine and mark the levels of lactates and acetyl-CoA. Histone lactylation is regulated by the level of both lactate and acetyl-CoA, which functions as the donor of the acetyl group and affects the level of acetylation.

### 2.4. Epigenetic Modification Regulates Mitochondrial Function

Metabolites related to mitochondria play important roles in epigenetic regulation as mentioned above, and in turn, epigenetic modification also influences the function of mitochondria.

DNA and histone methylation affect mitochondrial function via the regulation of SAM levels. SAM-regulated histone methylation has a positive effect on energy metabolism. The knockdown of NNMT, which catalyzes the methylation of nicotinamide with methyl from SAM, elevates SAM levels and increases H3K4 methylation in adipose tissue, and this enhances the key enzymes in the polyamine flux to increase systemic energy metabolism [273]. Maternal one-carbon metabolism related to SAM also affects the offspring’s energy metabolism [274]. Moreover, elevated lysine-specific demethylase LSD1 stimulates OXPHOS gene expression in NRF1 [275]. In addition, the differential DNA methylation patterns in skeletal muscle in obese and high-fat diet individuals alter the mitochondrial β-oxidation and mitochondrial number, while exercise training altering epigenetic modifications can improve mitochondrial function [276,277].

Histone acetylation has close relations with mitochondrial functions. HDAC inhibitors suppress oxidative stress and tissue damage caused by mitochondria dysfunction in mouse models [278]. HDA-1, the *C. elegans* ortholog of mammalian histone deacetylase (HDAC), is required for mitochondrial-stress-mediated activation of mitochondrial unfolded protein response, and it modulates mitochondrial stress response and longevity [279]. HDAC3 is essential for shaping mitochondrial adaptations for IL-1β production in macrophages through non-histone deacetylation. Furthermore, the loss of histone deacetylase SIRT3 or SIRT6 can activate the key glycolytic enzymes and lead to an increase in aerobic glycolysis or Warburg effect [280]. ATP-citrate lyase (ACL) is the enzyme that converts glucose-derived citrate into acetyl-CoA, and histone acetylation level depends on ACL [164]. However, conversely, histone acetylation also stabilizes ACL to promote tumor growth. When the acetylation is restrained, the ACL is inhibited at both the transcriptional and post-translational levels [281,282].

Different from methylation and acetylation, succinylation causes larger mass change and alters a positively charged side chain into a negatively charged one, thus causing a two-unit charge shift in the modified residues. Succinylation alters the rates of enzymes and pathways, especially mitochondrial metabolic pathways. For example, several SIRT5-targeted lysine residues related to the SDH subunit and succinylation at these sites may disrupt Complex II subunit–subunit interactions and electron transfer [283]. Succinylation also suppresses SDH activity. SIRT5-KO liver exhibits a compromised Complex II and ATP synthesis function in vivo [284], and SIRT5 KO reduces fatty acid oxidation and accumulation of long-chain fatty acyl-CoAs, as 60% of all proteins in fatty acid metabolism are succinylated [224]. This demonstrates that SIRT5 is a promoter of mitochondrial energy metabolism.

Other modifications of histones such as phosphorylation, sumolyation, and ubiquitination also have an extensive and dynamic interplay with mitochondria. The dysregulation of the balance between them and mitochondria leads to the disorder of metabolism and finally cellular homeostasis.

Mitochondrial regulation on epigenetics not only can occur via mitochondria-regulated metabolites, but also might occur via aerobic glycolysis (Warburg effect), which is tightly associated with mitochondrial OXPHOS activity. Inhibition of OXPHOS or mitochondrial synthesis may lead to the Warburg effect, and tumor cells switch their energy production from OXPHOS to glycolysis when mitochondrial energy is insufficient [285]. The Warburg effect leads to the increased production of lactate in cancer, and the lactate converted from pyruvate in tumor cells is the precursor for histone lactylation [261], and thus the Warburg effect regulates histone lactylation. Whether mitochondria have an effect on other epigenetics through the Warburg effect deserves further investigation. In turn, epigenetics also regulates the Warburg effect. For example, DNA methylation regulates the enzymes of glycolysis and the mitochondria dysfunction caused by aberrant DNA methylation triggers aerobic glycolysis [286,287]. M2 isoform of pyruvate kinase (PKM2), a key rate-limiting enzyme, contributes to the metabolism shift from OXPHOS to aerobic glycolysis, and the acetylation of PKM2 is stimulated by high glucose and thus promotes the Warburg effect.

HAT transfers acetyl group from acetyl-CoA to histone, leading to histone acetylation. HDACs, including the NAD+-dependent deacetylases sirtuins, remove the acetyl group from histones, and thus histone deacetylation occurs. The TCA intermediate metabolites in mitochondria such as acetyl-CoA and NAD+ regulate the acetylation. HMT transfers the methyl group from SAM to the histone, leading to histone methylation. Demethyltransferases such as LSD1 and JHDM remove the methyl group from histone, and thus histone demethylation occurs. JHDM requires FAD as a cofactor. Histone succinylation refers to the addition of succinyl group transferred from succinyl-CoA to histone. Histone succinyltransferase such as KAT catalyzes the succinylation of histone. Sirt5 located in mitochondria is a desuccinylase. O-GlcNAcylation is regulated by OGT and OAT. OGT utilizes UDP-GlcNAc to catalyze the addition of O-GlcNAc onto target proteins. Acetyl-CoA is one of the precursors of UDP-GlcNAc. Histone lactylation depends on intracellular lactate levels, and lactate and acetyl-CoA are the key metabolites to bridge the glycolytic switch. The IDH mutation also leads to an increase of succinate, fumarate, and 2-HG, and inhibits JHDM activity.

## 3. Epigenetics in Normal Development and Tumorigenesis

Cell fate is determined by the genes expressed, which are mainly regulated by epigenetic modifications of the genome. Epigenetic regulatory mechanisms are heritable changes in a sequence of stable but reversible gene function modifications yet do not affect the DNA primary sequence [288]. Epigenetics is one of the primary mechanisms causing early programming of cell proliferation, differentiation, death, and disease [289]. Epigenetics plays a key role in normal development such as embryonic development and stem cell differentiation. Its dysregulation often leads to diseases, including cancer.

Two models have been proposed to account for tumorigenesis: the clonal evolution model and cancer stem cell (CSC) model [290,291,292,293,294,295,296]. CSCs generate cellular heterogeneity, harboring stem-cell-like properties such as self-renewal and aberrant differentiation potential, and the differential hierarchy can be reversible under certain conditions [297,298,299]. Both genetic and epigenetic alterations are attributed to the stemness features installed in cancer cells. Cancer initiation and progression induced by oncogenic mutations are accompanied by significant epigenetic alterations. Epigenetic dynamics such as DNA methylation and histone modification patterns and nucleosome remodeling regulate cancer cell state [300,301]. Epigenetic modifications are associated with the silencing of tumor suppressor and differentiation genes in various cancers, and they can also deregulate fundamental signaling pathways controlling self-renewal and differentiation, including Wnt, Notch, Myc, and Hedgehog pathways [302]. Besides stem cells, epigenetics also regulates other cells: epigenetic pathways functioning in T cell fate stability and plasticity have been reviewed [303]. In addition, epigenetic regulation on gene expression controls innate immune cell differentiation, activation, and function [304,305].

Cell fate determination requires faithful execution of gene expression programs affected by alteration in genome and epigenome, which are increasingly recognized to respond to metabolic inputs. Metabolites such as acetyl-CoA, α-KG, SAM, NAD^+^, and O-GlcNAc play crucial roles in cell fate determination by the regulation of epigenetic modifications and gene expression.

### 3.1. DNA Methylation in Cell Development and Tumorigenesis

Most cell types display relatively stable DNA methylation patterns with up to 80% of CpG islands being methylated [306]. DNA methylation is essential for tissue development and aberrant DNA methylation contributes to tumorigenesis. DNA methylation controls chromatin conformation together with other histone modifications [307].

#### 3.1.1. DNA Methylation in Normal Development

DNA methylation plays important roles in physiology, including stabilizing cell-fate decisions, cell development, cell differentiation, and cell signaling [308,309,310,311]. It functions as a defense mechanism in preventing genomic instability due to transposon movements or the insertion of endoparasitic sequences in the genome. In females, DNA methylation is involved in the regulation of housekeeping genes and tissue-specific genes, and it can silence one allele of imprinted genes and compensate for the extra copy of the X chromosome [307,312]. The methylation level of the genome of oocytes is less than that of spermatozoa [313]. DNA methylation on CpG island in lineage-specific genes is essential for cell stemness and differentiation in human female hematopoietic stem/progenitor cells (HSPCs) [307,314].

In undifferentiated human embryonic stem cells (ETS), the promoter of key transcription factors OCT4 and NANOG genes are unmethylated or in a hypomethylation state [315], but they are undergoing rapid DNA methylation on differentiation in vitro and after implantation in vivo [315,316,317]. The expression of stem cell transcription factors such as OCT4, NANOG, and SOX2 is closely associated with the self-renewal of stem cells [318,319], and their expression is controlled by their promoter methylation level. The partial or full methylation of these pluripotency-associated genes leads to the downregulation of their expression and differentiation of ES cells [317,320,321]. Human ETS has unique CpG methylation characteristics, and it works together with histone modifications to drive stem cell differentiation [322,323]. The lack of DNA methylation in stem cells is associated with the differentiation ability, especially in the activation of cell-specific genes [38]. Similar to ES cells, DNA methylation is also crucial for the differentiation and maturation of the mammalian central nervous system (CNS) [324]. In fibroblasts, hundreds of DMGs involved in pluripotency and differentiation have been identified as being associated with lower transcriptional activity [325].

#### 3.1.2. DNA Methylation in Tumorigenesis

DNA methylation is the most extensively studied epigenetic mark for human diseases [326]. Aberrant DNA methylation has been associated with pathological gene expressions in a variety of human cancers [327]. Hypermethylation leads to the inactivation of certain tumor-suppressor genes in its promotor regions, thus resulting in gene silence in different cancer types [59] (Figure 3A). The changes in DNA methylation and differentially methylated regions (DMRs) have occurred in normal development and diseases [328,329]. Different DMR classes display tissue-specific DMRs [330,331], cell-type-specific DMRs, and DMRs arising in tumor heterogeneity [332,333,334,335,336]. Different classes of CpG islands are subject to different modes of regulation [337]. The alternative promoters have been frequently used by tumors to increase the isoform diversity and to activate the oncogenes that are repressed normally, thus changing the tumor status and evading host immune attacks by immunoediting [338,339,340,341]. Pediatric tumors are more prone to mutations and using epigenetic deregulation to promote tumorigenesis and progression compared with cancers in adults [342,343,344]. A high methylation level with decreased TET1 activity has been found to increase glioblastoma tumorigenesis and malignancy [345,346].

DNA methylation also regulates the neurodegenerative process. The dysregulation of 5hmC during brain development, including different 5hmC concentrations, affects gene regulation in neurodegenerative processes, and the reduction in TET activity in parallel with the reduction of 5hmC concentrations promotes neurodegeneration in the mouse model of Alzheimer’s disease [347,348,349,350].

#### 3.1.3. DNA Methylation in Cancer Stem Cell

DNA methylation includes hypermethylation and hypomethylation. Generally, hypermethylation of promoter DNA indicates that the gene is silenced, whereas hypomethylation indicates that the gene is activated [351,352,353]. DNA hypermethylation of tumor suppressor genes and global hypomethylation of oncogenes contribute to the formation of cancer stem cells and cancer cells’ drug resistance [354]. Hypermethylation of the LDHA gene promoter silences its expression in IDH-mutant-derived brain tumor stem cells (BTSCs) [355]. DNA methylation can be reduced by mutant KRAS-mediated glutaminolysis to increase Wnt signaling pathway, thus increasing cancer cell stemness and drug resistance in colorectal cancer [356]. In addition, breast CSCs have specific DNA methylation signatures that distinguish them from their non-BCSC counterparts [357].

The expression of DNMTs plays an important role in cancer stem cell development. Genetic mutations of DNMTs lead to epigenetic alterations that control cellular fate. For example, mutation of DNMT3A accompanied by mutation of KRAS supports self-renewal and induces malignancy in AML, which shows the synergistic effect between genetic and epigenetic alteration in initiating self-renewal proliferation of CSC population and malignancy [358,359]. DNMT1 promotes EMT to CSC in triple-negative breast CSCs. The inhibition of DNMT1 suppresses triple-negative breast CSCs populations [360]. In liver cancer, DNMT1-mediated methylation of BEX1 promoter leads to different stemness of cells in hepatoblastoma (HB) and hepatocellular carcinoma (HCC) and a subtype of HCC with high CSC features (CSC-HCC) [361]. The ubiquitin-like PHD and RING finger domains 1 (UHRF1) factor recruits DNMT1 to preserve the status of DNA methylation during DNA replication, and the UHRF1-DNMT1 complex is crucial for preserving genomic imprinting and overall epigenetic memory during cell divisions. Their activity is impaired upon cellular transformations into malignant states [362,363,364], and UHRF1 suppression promotes cancer stem cell differentiation to reduce metastasis [365].

DNA demethylation enzymes TET1 and TET2 not only promote somatic cells reprogramming into iPSCs but also play important roles in cancer stem cells. TET2 is essential for hematopoietic stem cell (HSC) differentiation and lineage commitment and is frequently mutated in myeloid malignancies, and its inhibition contributes to HSC expansion in aplastic anemia. The level of 5hmC has been demonstrated as an independent prognostic marker for renal cancer and is intimately associated with its overall survival [366]. The low expression of TET1 and 5hmC inhibited by Fisetin decreased proliferation and invasion of renal cancer stem cells [353]. Inhibition of TET2 and the corresponding hypermethylation enhanced the stem cell signature of AML stem cells. In glioblastoma, the CSCs possess increased levels of 5fC and 5caC [300,367]. 

TET-mediated DNA oxidation is highly dependent on metabolic states that reprogram the epigenome to maintain the stemness and promote tumor growth and metastasis [300] (Figure 3A). Various metabolites affect TET activity. Succinate and α-KG serve as key regulators of self-renewal and differentiation through the modulation of TET enzymes. Succinate, an important metabolite in the TCA cycle, serves as a mitochondrial retrograde signal that exerts effects on the epigenome by inhibiting histone demethylases and TET family activity [90,93]. Succinate accumulation links to mitochondrial MnSOD depletion via SDH/succinate/TET axis, thus leading to aberrant nuclear DNA methylation and altered cell fate [368]. α-KG is also important for the facilitation of epigenetic remodeling during premalignant cells, in which p53 has been activated [369]. Inhibition or mutation of SDH results in the accumulation of succinate and inhibition of α-KG-dependent enzymes, as mentioned above [92,93,370,371].

### 3.2. Histone Acetylation and Methylation in Normal Development and Tumorigenesis

#### 3.2.1. Histone Acetylation and Methylation in Normal Development

Histone acetylation plays important roles in biological processes that are involved in cell proliferation and differentiation [289]. Histone acetylation at transcriptional enhancers reflects global cell-type-specific expression [372]. Pluripotent embryonic stem cells (ES) exhibit chromatin domains with both transcriptionally active and silent histone modifications [373]. In the early differentiation period, gene transcription is regulated by histone modifications during their temporal transcription [374]. Pluripotent embryonic stem cells (PSCs) require acetyl-CoA and histone acetylation for the maintenance of pluripotency [375]. The differentiation of PSCs leads to a metabolic switch from glycolysis to OXPHOS, which in turn promotes their differentiation since inhibition of glycolysis leads to loss of histone acetylation and the pluripotent state [375,376,377].

HATs stimulate transcription through the acetylation of histones, thus leading to the relaxation of nucleosomes [378]. HATs and HDACs transmit differentiation signals to initiate appropriate epigenetic modifications, including the establishment of new histone modification or erasure of existing chromatin structure in in vitro differentiation of ES cells [379]. H3K9ac and H3K9me3 are almost undetectable in pluripotent ES cells and dramatically increase when cells leave the undifferentiated state in a short time [380]. H3K9me3 level increased within 2 days, corresponding to the deacetylation of this residue [381]. Maternal H3K27me3 controls DNA methylation-independent imprinting, and it is a hallmark of facultative heterochromatin that maintains transcriptional repression established during early development in many eukaryotes, allowing genes to encode specific transcription factors or early differentiation factors [373,382,383]. When the expression of pluripotent genes such as Oct4, Nanog, and Foxd3 is elevated, the genes’ chromatins are opened in stem cells and condensed after differentiation. The key enzymes of histone acetylation, KATs, directly acetylate several key transcription factors such as p53, FOXO1, and C/EBPα to modulate the transcription activity and affect cell homeostasis [300,384,385,386]. Besides ES, epigenetics also plays important roles in somatic stem cells, and HDACs, especially HDAC1 and HDAC2 expression levels, decrease significantly upon tissue differentiation, while HDAC3 and DNMT1 keep the same expression [387,388,389].

Histone acetylation and methylation affect each other; H3K4me1 and H3K27ac affect each other in different ways and play direct roles in nucleosome eviction and mRNA transcription at enhancers, respectively [390]. ES cell chromatin is in a highly dynamic state, with global DNA hypomethylation and a general abundance of transcriptionally active chromatin marks such as H3K4me3 and acetylation of histone H4, which leads to relatively relaxed chromatin of ES cells [38,391].

#### 3.2.2. Histone Acetylation and Methylation in Tumorigenesis

Histone lysine acetylation positively regulates gene transcription by promoting DNA accessibility via creating relaxation of chromatin structure and a more conducive environment for transcription, and the malfunction of HATs often perturbs appropriate gene-expression programs, thus leading to the development of disease [392] (Figure 3B).

Histone methylation and acetylation levels are closely associated with tumor initiation and growth. H3K27me3 induces gene silencing and regulates cell differentiation, and its dysregulation frequently happens in tumorigenesis [393,394]. Histone methylation reader protein ZMYND11 interacts with histone H3.3K36me3, thus leading to the suppression of tumor growth [395]. The epigenetic mark H4K16ac of H4 is prone to higher acetylation with age in healthy cells, while it tends to disappear in regions of the genome near age- and AD-dependent genes [396].

Histone deacetylase plays an important role in tumorigenesis (Figure 3B). The increased activity of HDAC2 and the decrease of H3K18/K23 acetylation have been found in AD [397,398]. KDACs, the erasers of acetylation, play a critical role in oncogenesis, and the inactivation of HDAC1 enhanced DOT1 activity and lymphomagenesis in mice [399]. Another typical example of epigenetics in modulating CSC properties is the epithelial-mesenchymal transition (EMT) process. ZEB1 is the key transcription factor of EMT, and it drives non-CSC to CSCs. Histone acetylation of the ZEB1 poises its chromatin and promotes EMT’s acquisition of CSC features in breast cancer [400,401,402,403]. Hypoxia inhibits the oxygen-dependent H3K27me3 demethylases and increases ZEB1 expression and prompts EMT’s acquisition of a CSCs phenotype [403].

The deacetylase sirtuin family plays an important role in cancer stem cells (Figure 3B). The NAD^+^-dependent deacetylase sirtuin family members, especially SIRT3, SIRT4, and SIRT5 located in mitochondria, play important roles in epigenetics such as deacetylation, demalonylation, and desuccinylation, thus affecting various aspects of cell fate [404].

Sirt3, the mitochondrial antioxidant, deacetylates FOXO3 and upregulates antioxidant enzymes such as manganese superoxide dismutase (MnSOD) to protect mitochondria against oxidative damage and maintain hematopoietic stem cells (HSCs) homeostasis [405,406,407]. The upregulation of Sirt3 expression improves their regenerative capacity in aged HSCs [408]. SIRT3 is highly upregulated in colon cancer and correlates with metastasis and tumor stages [409]. The knockdown of SIRT3 in ovarian cancer enhances the migration and invasion towards liver metastasis, while the overexpression of SIRT3 suppresses metastasis dramatically [410].

SIRT4 regulates the catabolism of multiple nutrients [411], and the loss of SIRT4 promotes the self-renewal of breast cancer stem cells [412]. High expression of SIRT5 increased migration and proliferation in prostate cancer cells and renal cell carcinoma [413,414]. SIRT5 deacetylates lactate dehydrogenase B (LDHB) and promotes its enzymatic activity, thus generating additional protons and increasing autophagy and tumorigenesis in colorectal cancer, and the knockdown of SIRT5 reduces cancer cell proliferation [415]. A high level of SIRT5 desuccinylates SDH and reduces its activity promoting cancer cell proliferation, while SIRT5 silencing results in hyper-succinylation of SDHA and reactivates it, thus preventing the growth of cancer cells [414]. SIRT5 also desuccinylates serine hydroxymethyl transferase (SHMT2), and the inactivation of SIRT5 downregulates SHM2 succinylation and decreases its activity in serine catabolism and inhibits cancer cell growth [416]. Chemoresistant prostate cancer cells shift the metabolism from glycolysis to OXPHOS, and this chemoresistance is modified by all three mitochondria sirtuins [417,418,419,420]. SIRT6 plays important roles in the regulation of cancer cell metabolism and the promotion of cancer cell apoptosis [421,422,423,424]. SIRT6 directly binds to the CDC25A promoter and decreases the acetylation level of histone H3 lysine 9 to inhibit colorectal cancer stem cell proliferation [425].

#### 3.2.3. Mitochondrial Metabolites Contribute to Normal Development and Tumorigenesis via Regulation of Epigenetics

The cofactors participating in histone acetylation affect cell fate. For example, α-KG generated from glutaminolysis is important for the M2 activation of macrophages, and it orchestrates macrophage activation through metabolic and epigenetic reprogramming. Glutamine metabolism supports M2 activation by facilitating the demethylation of H3K27, which is a repressive epigenetic mark that prevents the expression of M2 marker genes [426,427]. Acetyl-CoA, serving as acetyl donors for acetylation and co-factors for deacetylase enzymes, is critical in determining both macrophage and dendritic cell differentiation and function [428,429]. The treatment with acetate (a precursor of acetyl-CoA) increases H3K9ac and H3K27ac and delays early differentiation in human PSCs and mESCs, suggesting that a decline in acetyl-CoA drives the early differentiation in PSCs by reducing histone acetylation [375,430]. Therefore, acetyl-CoA is necessary for differentiation via histone-mediated regulation of gene expression in adult stem cells [431].

The intracellular metabolite levels are subject to both cell-intrinsic metabolic pathway activity and factors in the extrinsic environment such as nutrient availability. For example, tissue-specific and systemic nutrient availability modulates stem cell fate via α-KG-dependent dioxygenases, and this links the extracellular cues and cell fate decisions [84].

## 4. Reagents and Drugs Targeting Mitochondria and Epigenetics for Cancer Treatment

### 4.1. Targeting Mitochondrial Metabolites and Enzymes in TCA in Cancer

Targeting mitochondria in cancer has been reviewed often [9,432,433,434,435,436,437], and multiple mitochondria-targeted drugs have been developed [438,439,440,441]. Metabolites and enzymes in TCA cycles have been demonstrated to be therapeutic targets in cancers.

The level of acetyl-CoA is treated as a key indicator of a cell’s metabolic state and a second messenger in cells [186], and its fluctuation is associated with gene expression via the global changes in histone acetylation, and it promotes the upregulation of cell-migration-and-adhesion-related genes in glioblastoma [442,443]. Furthermore, acetyl-CoA production by specific metabolites promotes cardiac repair after myocardial infarction by histone acetylation [444]. In addition, acetyl-CoA metabolism supports pancreatic tumorigenesis, and it is used in histone acetylation and in the mevalonate pathway facilitating cell plasticity and proliferation, which suggests potential to target these pathways [445]. Acetyl-CoA carboxylase (ACC) is an ATP-dependent carboxylation of acetyl-CoA, and this rate-limiting enzyme in de novo fatty acid synthesis has also been demonstrated as a cancer therapy target [446].

Citrate is a key metabolite that bridges many important metabolic pathways. Abnormalities of citrate-metabolizing enzymes such as citrate synthase enzyme and ACL happen in various cancers [447,448,449,450]. Targeting citrate and citrate enzymes is a novel therapeutic strategy in cancer treatment [451]. Isocitrate dehydrogenase (IDH) enzymes catalyze the oxidative decarboxylation of isocitrate and therefore play key roles in the Krebs cycle and cellular homeostasis. IDH1 and IDH2 catalyze the conversion of isocitrate to αKG in the cell cytoplasm and mitochondria, respectively. They have been shown to induce cancer stem cell differentiation successfully in AML patients, and the IDH mutants produce oncometabolite d-2-hydroxyglutarate that inhibits downstream 2-oxoglutarate-dependent epigenetic modifiers, thus resulting in differentiation arrest of blast cells [452]. IDH-mutated human malignancies include glioblastoma [211,453], AML [454,455], intrahepatic cholangiocarcinoma, chondrosarcoma, and thyroid carcinomas [456,457]. Enasidenib and ivosidenib are potent and selective inhibitors of mutant IDH2 and IDH1, respectively, and both drugs have been approved by the Food and Drug Administration (FDA) for the treatment of AML [78].

The metabolite α-KG is membrane-impermeable, and intracellular α-KG inhibits starvation-induced autophagy and it has no direct respiration-inhibitory effect [80]. α-KG is an effector of p53-mediated tumor suppression, and the elevated intracellular levels of succinate (a competitive inhibitor of αKG-dependent dioxygenases) can blunt p53-driven tumor suppression [369].

Succinate plays an important role in inflammatory, hypoxic, and metabolic signaling [458], and targeting succinate and succinate metabolism can be a potential therapeutic method in aortic aneurysm and dissection, and in ischemia/reperfusion injury [459,460]. Dimethyl fumarate (DMF), a derivative of the Krebs cycle intermediate fumarate, is an immunomodulatory drug used in patients with multiple sclerosis and psoriasis [461,462].

Malate is a standard component in fluid therapy within a wide range of medical applications [463], and sunitinib malate has been used in the treatment of urothelial cancer [464].

Malate dehydrogenase (MDH) catalyzes the reversible interconversion of malate and oxaloacetate and their transport [465], and it has unique pharmacological selective and specificity properties and can be used as the potential antiparasitic target for drug development [466]. For example, urine malate dehydrogenase 2 can be used as a biomarker for the early detection of non-small-cell lung cancer [467]. Oxaloacetate (OAA) is a competitive inhibitor of human lactate dehydrogenase A (LDHA) in cancer cells, and there is an inverse correlation between lactate and OAA concentration in cells from patients with myeloma and normal healthy individuals, which indicates the relevance of OAA as a regulator of the Warburg effect in rapidly proliferating cells in humans [468]. Oxaloacetate treatment has a neuroprotective effect in rodent models of seizure and neurodegeneration. In amyotrophic lateral sclerosis (ALS) mice model, oxaloacetate treatment evaluates the neuromuscular function and lifespan [469]. Oxaloacetic acid (OA) is an intermediate of citric acid and a potential natural anti-pigmentation agent, and it can inhibit tyrosinase [470].

### 4.2. Targeting Epigenetics in Cancer

#### 4.2.1. Targeting DNA Methylation

Targeting epigenetics in cancer has emerged as a promising anticancer strategy. The analysis of epigenetic changes in DNA may help to identify new druggable targets for the treatment of cancer [471,472]. DNA methylation profile alteration leads to DNA instability and triggers tumorigeneses. Hypermethylation of tumor suppressors or hypomethylation of oncogenes may arise in diseases such as cancers [473,474,475]. DNA-methylation-targeted drugs have been developed widely. Nucleoside-derived inhibitors 5′-AZA, decitabine, and zebularine were approved by the FDA for the treatment of various types of cancer [476,477,478].

**a**.
**Targeting DNMTs**


Since 5′-AZA is unstable and highly toxic to cells, far less toxic DNMT inhibitors were developed, including RG108, which is a non-cytidine analog successfully tested in human prostate cancer cells, breast cancer cells, and colon cancer cells [479,480,481,482]. Hydralazine and procainamide act as competitive inhibitors by preferentially binding to DNMT1 [483,484]. SGI-1027 and its regioisomer MC3343 cause the degradation of DNMT1 [485], suppression of growth of leukemia cell line, and, in osteosarcoma, patient-derived xenograft [486,487]. The combination uses of AH507 (JAK 1/2 inhibitor) and SGI-1027 have a synergistic effect on the suppression of survival and proliferation of cervical cancer cells [488]. GSK3685032, a potent first-in-class DNMT1-selective inhibitor with improved in vivo tolerability, yields superior tumor regression and survival mouse models of AML [489]. CM272, a G9a/DNMT1 inhibitor, inhibits cholangiocarcinoma tumoroids and xenograft growth significantly [490]. UHRF1-DNMT1 is responsible for DNA methylation in DNA replication, and UHRF1 links DNA methylation and histone modifications. The downregulation of UHRF1/DNMT1 is upstream of many cellular events, including G1 cell arrest, demethylation of p16, and apoptosis [491]. UHRF1-DNA binding domain has been proposed as a drug target for cancer therapy [492,493,494]. UHRF1/DNMT1 axis is also a regulator of senescence [495].

Besides small molecular inhibition of DNMTs, other strategies to inhibit DNMT1 include the use of antisense oligonucleotides and miRNAs such as MG98 to suppress renal cell carcinoma [496]. Epigallocatechin-3-gallate, a polyphenol flavonoid, is involved in both DNA demethylation and modifications of histone acetylation and methylation [497]. Procaine (a local anesthetic) is capable of reducing the methylation level of CpG island in some cancers such as hepatoma, breast cancer, and gastric cancer [498] by inhibiting the binding of DNMT1 to the target genes [485,499].

Besides inhibiting DNMTs to suppress DNA methylation, the activation of DNA methylation can also be a therapeutic strategy. The DNMT activator budesonide supplementation increases DNA methylation and decreases tumor size [500,501].

**b**.
**Targeting demethylases**


DNA demethylation, as a potential target for cancer epigenetic therapy, has been reviewed before [502].

The demethylase TET 2 is an α-KG- and Fe2^+^-dependent dioxygenase catalyzing the iterative oxidation of 5mC, and the mutation and dysregulation of TET2 contribute to the development of multiple hematological malignancies [503]. Hydroxymethylation of DNA leads to hypomethylation of DNA in lupus [504]. Increased expression of TET enzymes in monocytes and T cells leads to abnormal global DNA hydroxymethylation in early RA patients, and methotrexate treatment partly reduces the level of DNA hydroxymethylation in rheumatoid arthritis [505]. TET inhibition by AGI-5198 and HMS-101 compound leads to tumor growth suppression in glioblastoma and AML cells [506,507]. In all, modification of DNA methylation either by hypo- or hypermethylation is effective in different types of tumors.

#### 4.2.2. Targeting Histone Methylation

Targeting histone methylation has been reviewed [508]. Both histone methyltransferases and demethylases can be targets in cancer.

The disruptor of telomeric silencing 1-like (DOT1L), a histone H3K79 methyltransferase, is a target for many diseases [509]. It sensitizes RB cells to chemotherapeutic drugs by impairing the DNA damage response, thereby potentiating apoptosis [510]. Many inhibitors were developed to target DOT1L, and among them, EPZ-5676 has been in clinical trials, but the clinical result is modest, and long-time treatment leads to drug resistance in AML [511,512,513].

Histone demethylase LSDl, the first identified demethylase of histone methylation, belongs to the FAD-dependent family of monoamine oxidases [163]. The dysregulation and overexpression of LSD1 are hallmarks of a number of human diseases [514]. LSDl inhibitor CC-90011 induces the cancer stem cell to differentiation in AML and small-cell-lung cancer (SCLC) cell lines, with antitumor efficacy, and CC-90011 is currently in phase 2 trials in patients with first-line, extensive-stage SCLC [514].

#### 4.2.3. Targeting Histone Acetylation

Histone acetylation renders a more relaxed DNA structure to facilitate the binding of transcription factors and other proteins for gene expression. Histone acetylation enzymes, including HATs and HDACs, are potential drug targets in many diseases [515,516].

CBP/p300, functioning as histone acetyltransferases and transcriptional co-factors, represents an attractive target for various diseases, including malignant tumor cancers such as leukemia and prostate [517,518,519]. CBP/p300 inhibitors have been widely developed, including pyrazolone derivatives, oxazolidinedione, barbituric derivatives, etc. [518]. The treatment with p300/CBP inhibitor B029-2 specifically decreases H3K18Ac and H3K27Ac and displays significant antitumor effects on hepatocellular carcinoma (HCC) cells in vitro and in vivo [520]. Its bromodomain inhibitor CCS1477, identified by CellCentric, is the only CBP/p300 inhibitor currently in clinical trials for the treatment of hematological malignancies and prostate cancer [518,521,522].

Bromodomain and extra-terminal domain (BET) proteins, including BRD1, BRD2, BRD3, BRD4, and BRDt, act as “readers” of acetylated histones, and they can recognize H3K27ac through their bromodomains [523,524]. BET inhibition has been applied in both solid tumors and hematological malignancies [525]. Blocking the binding between BRD4 and chromatin by BET inhibitor JQ1 inhibits the proliferation and differentiation in BRD4-dependent tumors [526].

Many HDAC inhibitors have been developed and characterized in tumors, thus causing tumor cell differentiation and apoptosis and resulting in the inhibition of tumor growth in animals [527,528]. HDAC inhibitors have been applied in many clinical trials for hematologic and solid cancers [529]. Some of the inhibitors have been approved by FDA and targeting lactylation suggests novel onco-therapeutic opportunities [267]. Histone deacetylation (HDAC) inhibitor mocetinostat synergizes with gemcitabine to inhibit in vivo pancreatic xenograft growth by disrupting EMT and the CSC phenotype [530]. Inhibition of HDACs with trichostatin A and vorinostat promotes epithelial differentiation, thus disrupting tumor sphere formation and subcutaneous tumor xenograft growth [531]. Abexinostat is a potent and pan-HDAC inhibitor with a favorable pharmacokinetic profile and no expected drug-drug interactions with pazopanib [532].

#### 4.2.4. Targeting Histone Lactylation

Lactate-derived lysine lactylation in histones serves as a new type of histone mark [265]. Lactylation and lactate have been proposed for cancer treatment. Histone lactylation could drive oncogene expression and accelerate tumorigenesis by activating the m6A reader protein, YTHDF2, which is a novel potential histone lactylation target for treating ocular melanoma [265]. The function of lactylation and targeting lactylation in tumorigenesis and development deserve further investigation.

## 5. Conclusions and Perspectives

In this review, we have presented an extensive discussion on mitochondria function in tumorigenesis from an epigenetic perspective. Even the primary function of mitochondria in cellular metabolism is energy production, and the metabolites generated in this process contribute to tumorigenesis by influencing gene expression and cell signaling via epigenetic regulations [85]. The interconnected metabolites generated in the TCA cycle communicate to the epigenetic machinery by providing precursors for epigenetic modifications, including DNA methylation and post-translation modification of histones. Post-translation modification of histones regulates the interaction of histones with DNA and the ability to recruit chromatin remodeling complexes necessary for transcription, replication, recombination, repair, and mitosis. Epigenetic regulation is a dynamic process that allows a high degree of plasticity of cell fate in response to tumor microenvironment.

The key metabolic enzymes in the TCA cycle such as IDH, SDH, and FH are mutated in various cancers, and these enzymes and other metabolites or intermediates in mitochondria have been proven to be tumor targets. Inhibition of the key enzymes in epigenetics such as methyltransferases or histone acetylases/deacetylases can reactivate epigenetically silenced tumor suppressor genes and thus decrease tumor cell growth. Some of the inhibitors targeting cancer stemness lead to cancer stem cell differentiation and suppress its growth by changing cell fate. Much more research can be observed in studies on epigenetics, which plays important roles in human diseases and is demonstrated as therapeutic targets in the treatment and diagnosis of cancer and proliferative diseases. Our current understanding and summary are not comprehensive enough to include all kinds of epigenetics regulation, but we believe that the recognition of the importance of understanding tumorigenesis, combined with mitochondrial metabolites and epigenetics, would be helpful for the design of therapeutic interventions.

## Figures and Tables

**Figure 1 cells-11-02518-f001:**
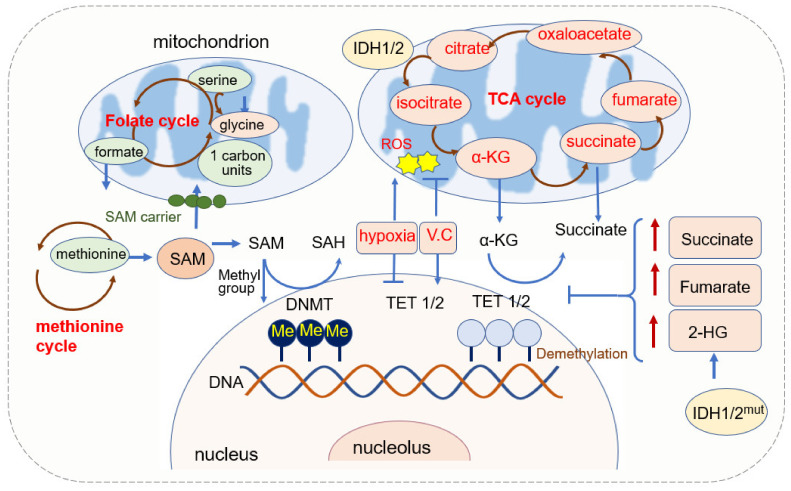
Mitochondrial function in DNA methylation.

**Figure 2 cells-11-02518-f002:**
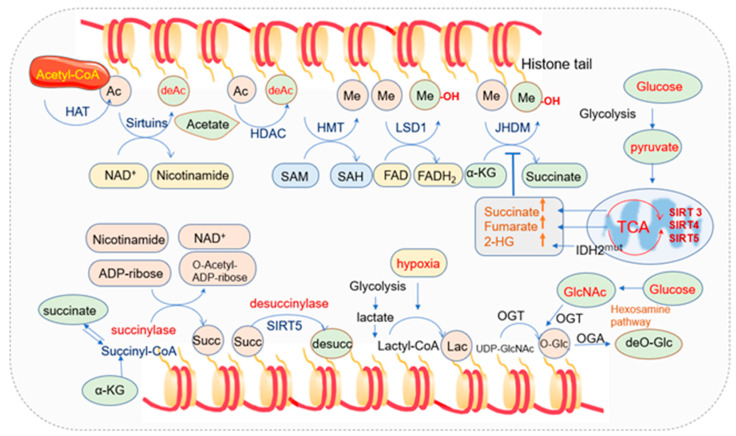
Mitochondria metabolites and histone post-translation modification.

**Figure 3 cells-11-02518-f003:**
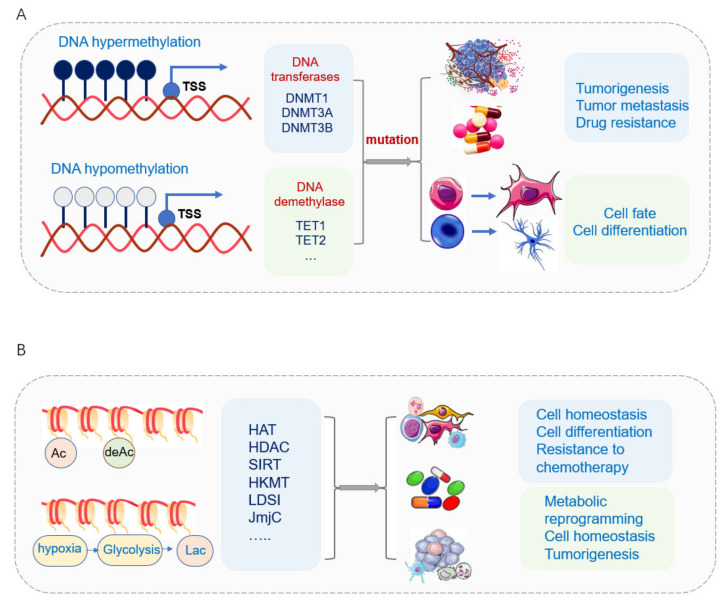
DNA methylation and histone modification affect normal development and tumorigenesis. (**A**) DNA hypermethylation of tumor suppressor genes and hypomethylation of oncogenes involved with tumorigenesis, tumor metastasis, and tumor drug resistance. DNA demethylation enzymes TET1 and TET2 play important roles in cancer stem cells. The loss of TET1/2 leads to hypermethylation and enhancement of cancer cell stemness. The status of DNA demethylases regulates CSC maintenance and differentiation. (**B**) The malfunction of HATs often perturbs the appropriate gene-expression program, leading to the development of disease. The key enzymes of histone acetylation KATs directly acetylate several important transcription factors to modulate the transcription activity and affect cell differentiation and cell homeostasis. The deacetylase sirtuin family affects cell fate via deacetylation, demalonylation, and desuccinylation. Histone acetylation is involved in CSC activity and drug resistance.

## Data Availability

Not applicable.

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
