# Peer review of "An Epigenetic Role of Mitochondria in Cancer"

_cells, 2022, doi:10.3390/cells11162518_

Round 1

Reviewer 1 Report

In the manuscript “An Epigenetic Role of Mitochondria in cancer”, Liu et al summarize the roles of mitochondria in key metabolites required for epigenetics modification and in cell fate regulation. They discuss the current strategies in cancer therapies via targeting epigenetic modifiers and related enzymes in metabolic regulation. The topic is interesting and attractive, and the manuscript is well-organized.  There are several suggestions to help the authors improve the manuscript.

Major

Epigenetics regulates the Warburg effect, and the latter will also affect the former. As the main place of the occurrence of the Warburg effect, the roles of mitochondria on epigenetics through the Warburg effect also need to be addressed.

In the “2.2. Mitochondria and RNA methylation” section, how about the relationship between mitochondria and RNA methylation? The authors should provide more progress has been made in this aspect. For example, 1. the effect of the metabolites from mitochondria on RNA methylation.  2. The roles of RNA methylation in mitochondria.

Minor

Lines 116-117, 355-369, 426-438, 452-455, the authors should make sure to use uniform, corresponding fonts, and sizes on the whole of the manuscript.

The full name of the abbreviations should be provided when it first appears. In addition, the abbreviation method needs to be unified, for example, CPG, CpG, CpGs.

The list of references does not meet the journal requirement. Before submitting a revision, be sure that your manuscript is properly prepared and formatted.

The Figures should be well-described in the figure legends. The elements in the figures need to be labeled.

In Figure 1, “oucleolus” should be “nucleolus”

In Figure 3, the two panels should be lined up exactly. The picture of the drugs is covered by others and led to image mutilation. The authors should modify those figures in their revision manuscript.

Author Response

Dear  Reviewer,

Thanks for your evaluation and great instructions. We’ve revised as per your request one by one. Please refer below answers in blue.

Open Review

REVIEWER 1#

Comments and Suggestions for Authors

In the manuscript “An Epigenetic Role of Mitochondria in cancer”, Liu et al summarize the roles of mitochondria in key metabolites required for epigenetics modification and in cell fate regulation. They discuss the current strategies in cancer therapies via targeting epigenetic modifiers and related enzymes in metabolic regulation. The topic is interesting and attractive, and the manuscript is well-organized.  There are several suggestions to help the authors improve the manuscript.

Major

Epigenetics regulates the Warburg effect, and the latter will also affect the former. As the main place of the occurrence of the Warburg effect, the roles of mitochondria on epigenetics through the Warburg effect also need to be addressed.

We thank the reviewer’s great suggestion. Warburg effect also known as aerobic glycolysis is tightly associated with mitochondrial function. As suggested, we added the roles of mitochondria on epigenetics through the Warburg effect at the end of part 2 on page 15.

In the “2.2. Mitochondria and RNA methylation” section, how about the relationship between mitochondria and RNA methylation? The authors should provide more progress has been made in this aspect. For example, 1. the effect of the metabolites from mitochondria on RNA methylation.  2. The roles of RNA methylation in mitochondria.

We agree with this review on this point and have expanded the role of mitochondria and RNA methylation by adding the discussion on the metabolites from mitochondria on RNA methylation and the roles of RNA methylation in mitochondria in parts 2.2.1 and 2.2.2. respectively.

Minor

Lines 116-117, 355-369, 426-438, 452-455, the authors should make sure to use uniform, corresponding fonts, and sizes on the whole of the manuscript.

We thank the reviewer for pointing out the format and we have changed to the same format of the manuscript.

The full name of the abbreviations should be provided when it first appears. In addition, the abbreviation method needs to be unified, for example, CPG, CpG, CpGs.

We thank the reviewer for bringing us attention to the abbreviations and we have carefully revised all of them.

The list of references does not meet the journal requirement. Before submitting a revision, be sure that your manuscript is properly prepared and formatted.

We thank the reviewer for pointing to the reference requests and we have revised the format as per request.

The Figures should be well-described in the figure legends. The elements in the figures need to be labeled. In Figure 1, “oucleolus” should be “nucleolus”

We thank the reviewer’s great instruction on figures and figure legends. We have added a detailed figure legend for each figure and corrected the mistake in spellings in Figure 1.

In Figure 3, the two panels should be lined up exactly. The picture of the drugs is covered by others and led to image mutilation. The authors should modify those figures in their revision manuscript.

We thank the reviewer for pointing out the problems with the figures and we have modified all the figures, which are much cleaner and clearer.

We would like to thank you again for evaluating our work. Please kindly let us know if additional information is required for you to judge our revised manuscript.

Sincerely Yours,

Yufeng Shi, Ph.D. Professor 

Tongji University Cancer Center,

Shanghai Tenth People’s Hospital,

School of Medicine, Tongji University,

Shanghai, China

Reviewer 2 Report

The review by Liu et al is a very thorough review of the factors associated with mitochondria that effect the genome in cancer and therefore have an epigenetic effect.

Major Comments:

This review needs serious editing for the English language.

There were several reviews of this nature in the literature.  I have listed 2 of them below.  Please explain why your review is different from these already published reviews

1.     Peng Y, Liu H, Liu J, Long J. Post-translational modifications on mitochondrial metabolic enzymes in cancer. Free Radic Biol Med. 2022 Feb 1;179:11-23. doi: 10.1016/j.freeradbiomed.2021.12.264. Epub 2021 Dec 17. PMID: 34929314.

2.     Sharma J, Kumari R, Bhargava A, Tiwari R, Mishra PK. Mitochondrial-induced Epigenetic Modifications: From Biology to Clinical Translation. Curr Pharm Des. 2021;27(2):159-176. doi: 10.2174/1381612826666200826165735. PMID: 32851956.

In the section starting at line 70 talking about metastasis, there is no context for this paragraph and the next paragraph doesn’t help either.   Please give a better explanation of how you are tying in metastatic events and mitochochondrial factors.

Minor Comments:

Even though there are many grammatical errors, the one that may not get caught is the spelling of Waddington on line 48.

Author Response

Dear Reviewer,

Thanks for your evaluation and great instructions. We’ve revised as per your request one by one. Please refer below answers in blue.

REVIEWER 2#

The review by Liu et al is a very thorough review of the factors associated with mitochondria that effect the genome in cancer and therefore have an epigenetic effect.

Major Comments:

This review needs serious editing for the English language.

We thank the reviewer for pointing out the language problems and we have carefully proofread it and modified the expressions to be clearer.

There were several reviews of this nature in the literature.  I have listed 2 of them below.  Please explain why your review is different from these already published reviews

1.Peng Y, Liu H, Liu J, Long J. Post-translational modifications on mitochondrial metabolic enzymes in cancer. Free Radic Biol Med. 2022 Feb 1;179:11-23. doi: 10.1016/j.freeradbiomed.2021.12.264. Epub 2021 Dec 17. PMID: 34929314.

  1. Sharma J, Kumari R, Bhargava A, Tiwari R, Mishra PK. Mitochondrial-induced Epigenetic Modifications: From Biology to Clinical Translation. Curr Pharm Des. 2021;27(2):159-176. doi: 10.2174/1381612826666200826165735. PMID: 32851956.

We thank the reviewer for bringing our attention to these two reviews. We noticed their existence and prepared ours with a different focus.

Peng et.al emphasized the important function of TCA enzymes and other metabolic enzymes in tumorigenesis and discussed post-translation modification on the proteins involved in mitochondria biogenesis and regulation. Sharma et.al Mainly focused on the maintenance of mitochondria itself including the mitochondria redox balance, mtDNA damage and repair, and the epigenetic regulation of mitochondrial functions. While our review mainly talks about the crosstalk between mitochondrial metabolites and epigenetics, we emphasized the mitochondrial importance in cancer from a perspective of epigenetics and targeting both mitochondria and epigenetics can be a novel promising therapeutic method. We cited both papers in our revised manuscript.

In the section starting at line 70 talking about metastasis, there is no context for this paragraph and the next paragraph doesn’t help either.   Please give a better explanation of how you are tying in metastatic events and mitochondrial factors.

We thank the reviewer for pointing out this, to make the whole content more focused and concise, we decide to delete the discussion of mitochondria and metastasis as it is another important field deserving higher attention. This manuscript mainly focuses on the epigenetic role of mitochondria in tumorigenesis.   

Minor Comments:

Even though there are many grammatical errors, the one that may not get caught is the spelling of Waddington on line 48.

 We thank the reviewer for pointing out this problem and we have carefully proofread the manuscripts and corrected them in the revised version.

We would like to thank you again for evaluating our work. Please kindly let us know if additional information is required for you to judge our revised manuscript.

Sincerely Yours,

Yufeng Shi, Ph.D. Professor 

Tongji University Cancer Center,

Shanghai Tenth People’s Hospital,

School of Medicine, Tongji University,

Shanghai, China

Round 2

Reviewer 2 Report

This looks fine, however the authors never corrected Waddington on line 48. It needs a 'g'.